# Genetics and Epigenetics of the X and Y Chromosomes in the Sexual Differentiation of the Brain

**DOI:** 10.3390/ijms232012288

**Published:** 2022-10-14

**Authors:** Lucas E. Cabrera Zapata, Luis Miguel Garcia-Segura, María Julia Cambiasso, Maria Angeles Arevalo

**Affiliations:** 1Instituto de Investigación Médica Mercedes y Martín Ferreyra (INIMEC), Consejo Nacional de Investigaciones Científicas y Técnicas (CONICET), Universidad Nacional de Córdoba, Córdoba 5016, Argentina; 2Instituto Cajal (IC), Consejo Superior de Investigaciones Científicas (CSIC), 28002 Madrid, Spain; 3Cátedra de Biología Celular, Facultad de Odontología, Universidad Nacional de Córdoba, Córdoba 5000, Argentina; 4Centro de Investigación Biomédica en Red de Fragilidad y Envejecimiento Saludable (CIBERFES), Instituto de Salud Carlos III, 28029 Madrid, Spain

**Keywords:** X chromosome, X-linked genes, Y chromosome, Y-linked genes, sex differences, Four Core Genotypes mouse, neurogenetics, neuroepigenetics, brain sexual differentiation

## Abstract

For many decades to date, neuroendocrinologists have delved into the key contribution of gonadal hormones to the generation of sex differences in the developing brain and the expression of sex-specific physiological and behavioral phenotypes in adulthood. However, it was not until recent years that the role of sex chromosomes in the matter started to be seriously explored and unveiled beyond gonadal determination. Now we know that the divergent evolutionary process suffered by X and Y chromosomes has determined that they now encode mostly dissimilar genetic information and are subject to different epigenetic regulations, characteristics that together contribute to generate sex differences between XX and XY cells/individuals from the zygote throughout life. Here we will review and discuss relevant data showing how particular X- and Y-linked genes and epigenetic mechanisms controlling their expression and inheritance are involved, along with or independently of gonadal hormones, in the generation of sex differences in the brain.

## 1. Sex and Brain: Not All about Gonadal Hormones

Sex differences in the brain, found at a wide range of levels from neuronal and glial functions to anatomical, physiological and behavioral phenotypes, are now understood to be the result of two major factors acting across the lifespan: (1) a sex-specific trophic environment due to differences in the secretion of gonadal hormones and (2) a distinct genetic and epigenetic pattern depending on sex, generated by differences in the expression of genes linked to the X and Y sex chromosomes. However, it took several decades of research to first identify these factors as generators of phenotypic sex differences in the brain and then to begin to comprehend how they act to shape sex-specific neural circuits, often interacting with each other and with the environment in complex ways. Although sex differences in the human brain can also be found and discussed in terms of sexual orientation and gender identity [1], with “female” and “male” being frequently used to define gender, this review focuses on addressing sex differences in the brain in terms of biological sex without discussing gender identity. Thus, to avoid misinterpretations, we have circumscribed here the definitions of “female” as an XX individual carrying ovaries and “male” as an XY individual carrying testes.

Towards the end of the 1950s, Charles H. Phoenix, Robert W. Goy, Arnold A. Gerall and William C. Young performed a series of experiments that would lay the groundwork for beginning to understand how gonadal hormones affect the developing brain to establish a sex-specific neural substrate. These authors reported for the first time how prenatal testosterone administration in rodents irreversibly affected mating sexual behavior in adulthood, masculinizing and defeminizing its expression in females [2]. In the following years, numerous articles were published pointing out the relevance of the perinatal action of gonadal steroids in the definition of adult reproductive behavior [3,4,5,6]. This important corpus of evidence was the basis for the formulation of the classical hypothesis of brain sexual differentiation, which holds that sex differences in the central nervous system are a direct consequence of gonadal steroids, postulating two types of actions for these hormones in the brain: organizational and activational. *Organizational actions* describe the effects that gonadal hormones have on shaping neural circuitry during gestational and early postnatal development, whereas *activational actions* refer to the physiological/behavioral responses of the sexually differentiated adult brain to sex steroids [7]. According to this hypothesis, during a perinatal sensitive/critical period (in rodents starting around embryonic day 18 —E18— and ending around postnatal day 10 —P10—), testosterone produced by the developing testes and its metabolites, dihydrotestosterone (DHT) and 17β-estradiol (E2) [8,9,10], actively organize male-type brain circuits, such that, in adulthood, these circuits respond to the activational effects of gonadal hormones by displaying typically male behaviors. In females, on the contrary, the absence of gonadal secretions during this period leads to the development of a female brain, which will be able to generate female-specific physiological/behavioral responses to the appropriate hormonal influence in adulthood [11,12,13,14].

With the organizational/activational hypothesis of brain sexual differentiation, the idea that differences associated with the X and Y chromosomes may contribute to the sex-dependent development of non-gonadal tissues lagged long behind. For years, the role of X and Y as factors causing sex differences in the brain were not properly addressed, mainly due to the enormous amount of experimental evidence that emerged during the second half of the 20th century pointing to sex hormones as the sole determinant of phenotypic sex differences in vertebrates and, importantly, because of the lack of experimental tools that would allow clear discernment of sex chromosome effects independently of gonadal secretions. However, over the last decades, with the development of genetic engineering and the generation of certain transgenic animals, it has been possible to identify numerous sex differences in the brain whose origin and development are either completely independent of sex hormones or cannot be explained solely through their action [15,16]. These partial or total gonad-independent sex differences are tightly controlled by genetics and epigenetics of sex chromosomes.

## 2. Sex Chromosome Complement and Brain Sex Differences

The concept of *sex determination* refers to the process by which it is defined, in early embryogenesis, to follow a female or male sexual differentiation pathway, which is directly controlled by the sex chromosome complement of the original zygote. In mammals, the alternative sex chromosome complements are XX (two copies of the X chromosome, one inherited from the father and the other from the mother) and XY (the X chromosome inherited from the mother and the Y chromosome inherited from the father), with XX individuals being determined as females and XY as males [17]. This is mainly due to the existence of the testis-determining *Sry* gene on the Y chromosome, which begins to be expressed in early embryogenesis (around E10.5 in mice) in the undifferentiated gonads of XY individuals to drive the process of testis differentiation. In XX individuals, certain autosomal and X-linked genes, which are silenced by *Sry* in XY embryos, induce the ovarian differentiation program [18,19].

Some of the earliest evidence for the development of sexually differentiated phenotypes independent of hormonal action was obtained from the study of cells/tissues isolated from embryos before the onset of gonadal secretion of sex hormones and their influence on brain organization, i.e., before the critical period of perinatal sensitivity [20,21,22]. For instance, Pilgrim’s laboratory showed in rats a greater number of dopaminergic neurons in mesencephalic cultures from females than from males, as well as a higher content of the neurotransmitter dopamine in neurons derived from diencephalon of females. Interestingly, treatment of cultures with E2 or testosterone did not eliminate sex differences. Since all neuronal cultures were established from E14 embryos (a developmental time well before the onset of the critical period in rats), these differences found between female and male dopaminergic neurons cannot be explained by the sexually dimorphic action of gonadal secretions [23,24].

Another set of evidence of great interest for its implications in the understanding of complex sex-specific behaviors in adult animals came from the study of birds. In zebra finches (*Taeniopygia guttata*), males produce a courtship song that females are unable to perform, the neural circuitry responsible for controlling this song being much larger and more elaborate in males [25]. Although it has been observed that estrogen treatment of newborn females leads to the development of a song due to masculinization of the brain regions that control it [26,27], and that blockade of these steroids in males prevents complete masculinization [28], these effects are always partial, often not observed at all, or require estrogen/androgen doses for masculinization of females much higher than physiological levels measured in males, making a purely hormonal explanation unlikely [29]. In contrast to mammals, male birds are homogametic (with a ZZ sex chromosome complement) and female birds are heterogametic (with a ZW complement). The discovery of a finch with bilateral gynandromorphism made it possible to evaluate, to some extent, the impact of the sex chromosome complement on the generation of male and female phenotypes. These “half-male, half-female” finches have a testis and male plumage on the right half of the body and an ovary and female plumage on the left half. By in situ hybridization, it was shown that W-linked genes were expressed almost exclusively on the left half of the brain, whereas Z-specific genes were expressed ubiquitously but at markedly higher levels on the right half of the brain, as when comparing ZW males with ZZ females. Furthermore, despite the fact that the whole brain developed in the same environment of circulating gonadal hormones, the brain regions governing singing were much larger on the right hemisphere than on the left hemisphere, an effect that can only be attributed to the difference in the sex chromosome complement of cells from one half of the brain and the other [30].

The development of certain transgenic mice has allowed a major leap in the study of sex chromosome complement as a determinant of phenotypic sex differences in the last two decades. One of these transgenic models is the Four Core Genotypes (FCG) mouse, which combines two mutations in the same murine line: a spontaneous deletion of the *Sry* gene from the Y chromosome, generating a “Y minus” or “Y^−^” [31], and the subsequent reinsertion of this gene into autosome 3 [32,33]. Thus, it was possible to dissociate the inheritance of the Y chromosome from that of the *Sry* gene and the subsequent differentiation of testes, generating mice whose gonadal phenotype (testes/ovaries) is independent of the sex chromosome complement (XY/XX). Mice that inherit an X chromosome, a Y^−^ chromosome and the *Sry* reinserted on chromosome 3 (XY^−^*Sry* mice), develop testes and in adulthood are fertile males. Breeding XY^−^*Sry* males with XX females makes it possible to obtain the four different genotypes that give the model its name: XX females (XXf), XY^−^ females (XYf), XY^−^*Sry* males (XYm) and XX*Sry* males (XXm). Comparison of a variable among these four groups allows independent assessment of the effects of the factors: (1) gonadal sex (testes/ovaries and the hormonal environment associated with each type of gonad), (2) genetic sex (XY/XX sex chromosome complement) and (3) the interaction between these two factors (Figure 1). The main advantage of the FCG model is that the sex chromosome effects can be measured even after gonadal differentiation, so that the assessment of the genetic sex role is not restricted to the stage prior to the perinatal critical period.

FCG mice have been widely used in recent years in the study of sexual differentiation of the brain, allowing for a clearer assessment of the specific contributions of gonadal hormones and sex chromosome complement. In the first published work using the murine model for this purpose, a sex difference in the proportion of dopaminergic neurons due to sex chromosomes was demonstrated in mesencephalic neuronal cultures from E14.5 FCG embryos: at both 6 and 11 days in vitro, cultures derived from XY embryos had significantly more dopaminergic neurons (immunoreactive for tyrosine hydroxylase) than those derived from XX, regardless of the gonadal sex of the donors [34]. In another series of studies using adult FCG animals, a higher density of vasopressin-immunoreactive fibers was reported in the lateral septum of XY compared to XX mice without effect of gonadal sex, observing, in turn, more aggressive and less parental care behaviors in gonadal male and XYf mice than in XXf mice, with evidence associating these behavioral sex differences with those found at the level of the vasopressinergic system [35,36]. In this case, although the reported sex differences depend directly on the sex chromosome complement, it is evident how the organizational/activational influence of testicular secretions masculinizes and defeminizes the behavior of XXm mice, indicating synergistic effects between the sex chromosome complement and the hormonal environment to sex-specifically differentiate the assessed behaviors. The effect of sex chromosomes in determining sexually dimorphic expression patterns for autosomal genes, including genes encoding transcription factors, enzymes and neurotransmitters, has been explored by some laboratories. A remarkable case is *Cyp19a1*, a gene located on chromosome 9 in mouse and coding for aromatase, the enzyme catalyzing testosterone conversion to E2 not only in the gonads but, importantly, in the brain [37,38]. Higher aromatase mRNA and protein expression in stria terminalis and anterior amygdala has been reported in XY than in XX E16 FCG mice. E2 or DHT stimulation of amygdala neuronal cultures derived from E15 embryos resulted in increased aromatase expression only in XX neurons, thus abolishing sex differences observed under control conditions [39,40,41]. Similarly, hippocampal neuronal cultures from E17 wild-type mice also showed greater aromatase mRNA levels in XY than in XX neurons [42]. Several other works using FCG mice have provided valuable evidence on sex chromosome complement involvement in determining sex differences in Alzheimer’s disease vulnerability [43], response to nociceptive stimuli [44], habit formation [45], autoimmune disease susceptibility [46], developmental neural tube defects [47], bradycardic baroreflex response [48], sodium appetite and renin-angiotensin system [49,50] and ethanol intake [51], among others [52].

## 3. Genetics and Epigenetics of the X and Y Chromosomes and Sexual Differentiation

In mammals, sex chromosomes have evolved under different selective pressures that generated an extreme and more than evident divergence between the two, not observed in any other homologous pair. It has been proposed that the mammalian X and Y chromosomes evolved from an ordinary autosomal pair that began to diverge at least 180 million years ago [53,54,55]. The first differentiating event would have been the acquisition of a testis-determining gene or region on the “proto-Y” autosome, from which the *Sry* gene would evolve. Subsequently, a series of large-scale inversion mutations occurred mainly on this chromosome, leading to a drastic restriction in its ability to recombine with the X during meiosis. Finally, most of the Y underwent genetic degeneration due to deletions and loss of entire genes, which together reduced it to minimal expression. In parallel and responding to this gradual reduction of the Y, a complex mechanism of wide transcriptional silencing of one of the two Xs in females, known as the X chromosome inactivation (XCI), began to develop as a process that compensates for the dosage imbalance of X-linked genes between XX and XY individuals [55,56,57].

As a result of this divergent evolution, the modern Y is considerably smaller than the X and contains only 48 known protein-coding genes in humans and 12 in mouse, most of which serve functions restricted to testicular determination and spermatogenesis [58]. Some of these genes reside in the small pseudoautosomal regions (PARs) located at the ends of both X and Y chromosomes. PARs share sequence identity since they recombine as autosomal regions during meiosis [59]. While humans have two PARs per sex chromosome, PAR1 on Xp/Yp and PAR2 on Xq/Yq, each containing only about 15 and 4 genes, respectively, mice carry a single PAR with two genes only [60,61,62,63]. Therefore, most genes on the X and Y chromosomes are located in the regions between the PARs, which are the male specific region of the Y (MSY) and the non-PAR of the X (Figure 2). Although MSY genes cannot recombine with X and are male-specific, some of them still retain an X-linked counterpart with which they share some level of remaining homology and are ubiquitously expressed in the organism [56,64]. These X/Y homologs are highly dosage-sensitive and conserved among mammals. This suggests that deleterious effects due to the loss of the Y-linked copy during Y degeneration and the subsequent decreased expression probably acted as selective pressures favoring the conservation of one copy on each sex chromosome for these loci [65,66,67]. Some important X/Y homologs include *Kdm6a*-*Utx*/*Kdm6c*-*Uty*, *Kdm5c*/*Kdm5d*, *Ddx3x*/*Ddx3y*, *Usp9x*/*Usp9y*, and *Eif2s3x*/*Eif2s3y*.

The X, on the other hand, is much larger than the Y and contains more than 1000 genes, of which more than 800 code for proteins in humans and mice [58]. As mentioned above, the existence of two Xs in XX individuals and only one in XY individuals causes a dosage imbalance in the copy number of virtually all of the X-linked genes between the sexes, with the exception of those encoded on PARs. This imbalance is compensated during early embryogenesis by XCI, which silences transcription of one of the two Xs in each cell of the XX blastocysts. Thus, XCI defines a transcriptionally inactive X (Xi) and a transcriptionally active X (Xa) that will be clonally inherited by mitosis to all the cells eventually shaping an XX organism [64,68]. The molecular mechanism by which XCI occurs is extremely complex and includes the selective expression from and physical coating of the future Xi of multiple copies of the *Xist* (*X-inactive specific transcript*) long noncoding RNA, as well as the accumulation of repressive methylation marks on histones and DNA, all of which contribute to the heterochromatic conformation of the silenced X [69,70,71,72]. Since XX individuals inherit one X from the father and the other from the mother and XCI in eutherian mammals *randomly* inactivates one of the two Xs in each of the embryonic cells, the ultimate result is a mosaic phenotype or mosaicism for the expression of heterozygous loci, with some patches of cells expressing the paternal and others the maternal X genes [64,68,73].

However, despite the XCI mechanism, imbalances in the expression levels of X-linked genes can still occur between males and females. Firstly, it is now known that all chromosomes received by a new individual are inherited carrying a number of epigenetic marks that will condition their expression and that are different depending on whether the paternal or maternal chromosome is considered for each homologous pair, a phenomenon called genomic imprinting [74]. Thus, some X-linked genes could be expressed to a greater or lesser extent depending on whether the allele is inherited from the father or the mother, so that, in XX individuals, silencing one or the other X in each cell would not have exactly the same effect. Moreover, in XY individuals the X is always inherited from the mother, which means that for these individuals the imprinting of the X is always maternal [75,76,77,78]. Secondly, XCI does not result in complete repression of all Xi genes, as some “escape” inactivation and can thus be expressed from both the Xa and the Xi [79]. In humans, about 23% of X-linked genes are consistently expressed from both copies in XX cells, so their expression levels are significantly higher in these cells compared to XY cells [80,81]. In mice, about 7% of X-linked genes escape inactivation [82]. Notably, while some X genes consistently escape XCI in all species, individuals and tissues studied, many others vary significantly in the extent of their escape depending on species, individuals of the same species, developmental stage, cell or tissue type and health/disease condition [83,84,85,86].

Therefore, considering the particular nature of sex chromosomes and their inheritance, it is possible to identify at least five mechanisms potentially involved in the establishment of sex differences independently of the action of gonadal hormones (Figure 2): (1) expression of MSY-linked genes (only present in XY individuals), (2) increased expression in XX individuals of X-linked genes escaping XCI, (3) reduced expression of PARs-linked genes in XX individuals due to XCI, (4) differences in X gene expression due to genomic imprinting and (5) sequestration of transcriptional regulators such as heterochromatin assembly factors and other epigenetic modifiers for Xi silencing in XX individuals, which would limit the availability of such factors for regulation of autosomal gene expression [15,58,81,87,88]. Delving into these differentiation mechanisms is contributing to the final understanding of sex chromosome dosage effects on genome-wide autosomal expression, which are already known to span a diversity of cellular functions such as cell fate, cell-cycle regulation, chromatin organization, immune response signaling, protein trafficking and energy metabolism [58,65,89,90,91].

## 4. X-Linked Genes and Sexual Differentiation of the Brain

Although the X chromosome does not comprise more than 5% of the human and mouse genomes, it exhibits an interesting characteristic for neurobiology: a unique enrichment in the number of brain-relevant genes, containing nearly a sixfold greater number of genes involved in neurodevelopmental and neurophysiological processes than autosomes [92,93,94]. Furthermore, studies assessing mRNA levels clearly show that X-linked genes are expressed at higher levels in the nervous system than in other tissues, both in mice and humans [94,95,96]. Mutations in several of these genes cause different disorders characterized by cognitive impairment and commonly referred to as X-linked mental retardation (XLMR) conditions, with Rett syndrome, Fragile X syndrome and Börjeson-Forssman–Lehmann syndrome being just a few [97,98,99].

Among the numerous X-linked genes involved in neurodevelopment and neurophysiology, we can highlight, for example, *Kdm6a*, *Kdm5c*, *Mecp2*, *Hdac8*, *Morf4l2*, *Msl3*, *Phf6*, *Ddx3x*, *Eif2s3x*, *Fmr1*, *Usp9x*, *Mid1*, *Ogt*, *Syp* and *Tmem47*. Several of these genes encode epigenetic regulators of transcription. This is the case of the histone demethylases KDM6A and KDM5C [100], the methylated DNA-binding protein MECP2 [101], the histone deacetylase HDAC8 [102], the component MSL3 of the MSL acetyltransferase complex [103] and the component MORF4L2 of the NuA4 histone acetyltransferase and mSin3a histone deacetylase complexes [104]. Others encode transcription factors such as PHF6 [105], proteins involved in translation and mRNA metabolism such as DDX3X, EIF2S3X and FMR1 [106,107,108], modulators of protein activity through posttranslational modifications such us deubiquitinase USP9X, ubiquitin ligase MID1 and glycosyltransferase OGT [109,110,111], or integral membrane proteins such us TMEM47/BCMP1 and the synaptic vesicle-protein synaptophysin (SYP) [112,113]. All these X-linked genes are most likely involved in the development of sex differences in the brain, making their study at that level extremely interesting [88,114]. Notably, most of the X genes actively involved in the sexual differentiation of the brain belong to the very small group of the X/Y homologs (e.g., *Kdm6a*, *Kdm5c*, *Ddx3x*), with X homologs having evolved the pattern of escape from XCI and commonly expressing higher in XX than in XY individuals. Here we will review important evidence emerging especially in the last 5 years that points to some of these genes as critical factors in brain sexual differentiation.

*Kdm6a* (or *Utx*) encodes a histone demethylase enzyme that promotes chromatin accessibility by removing repressive H3K27 methylation [115,116]. It is a non-PAR X-linked gene and retains a Y homolog, *Kdm6c* or *Uty*, although in vivo demethylase activity of this Y counterpart is almost lost and its physiological role as an H3K27 demethylase is currently under discussion [117,118]. KDM6A not only promotes gene expression by catalyzing H3K27 demethylation, but also through demethylation-independent functions by interacting with H3K4 methylation complexes and H3K27 acetyltransferases [119,120,121,122]. Thus, KDM6A is a key epigenetic modifier promoting chromatin accessibility and transcription genome wide. In XX individuals, *Kdm6a* consistently escapes XCI in different mouse and human tissues, including the brain, resulting in higher KDM6A expression in XX than in XY individuals [81,82,123,124,125].

In the nervous system, *Kdm6a* is involved in the determination of neural stem cells and their subsequent differentiation into glial cells and neurons [122,126,127,128,129]. *HOX* genes are a major target of KDM6A during cell differentiation, with the demethylase being required for the removal of H3K27me3 at promoter regions and subsequent expression of HOXB1, HOXD10, HOXD11 and HOXD12 [130,131]. During retinoic acid-induced neuronal differentiation of human embryonic stem cells in vitro, preferential recruitment of KDM6A for H3K27 demethylation and transcriptional activation of HOX genes of clusters A, B, C and D has been reported [132]. *Kdm6a* deletion during brain development leads to severe defects in multiple areas including cerebral cortex, hippocampus and hypothalamus, often with different effects for males and females. For instance, whereas homozygous knockout of *Kdm6a* is fatal in female mice around 11–12 days of embryonic development due to severe heart malformations and neural tube closure defects, knockout male mice survive to adulthood and are fertile, suggesting that the presence of Y-linked *Uty* in males is partially preventing the deleterious effects of *Kdm6a* loss [117,133]. *Kdm6a* deficiency by knockdown and conditional knockout in the developing cerebral cortex increased neural stem cells proliferation in basal layers (ventricular and subventricular zones) and decreased the number of differentiated neurons in the upper layer (cortical plate) at E16.5/E17.5, these effects being less significant in males than in females [134]. Regarding the hippocampus, *Kdm6a* conditional deletion during embryonic development in mice resulted in repression of autosomal genes involved in neuritogenesis and synaptogenesis, impaired dendritic maturation, functional alterations in synaptic plasticity, deficits in spatial learning and increased anxiety-like behaviors [135]. Remarkably, such studies were conducted using only *Kdm6a* knockout males, as virtually all knockout females died within 3 weeks after birth [135].

The presence of two functional copies of *Kdm6a* in XX versus just one copy in XY embryos is required for the expression of early sex-specific phenotypes in the developing hypothalamus before gonadal hormones organize the brain. In FCG mice, hypothalamic neurons carrying two X chromosomes showed higher *Kdm6a* mRNA levels than XY neurons, regardless of gonadal sex. Remarkably, this XX > XY expression pattern was not affected by different hormonal conditions between embryonic (E14) and postnatal (P0, P60) ages or by the treatment with E2 of hypothalamic neurons in vitro, indicating that sex differences in *Kdm6a* expression are directly dependent on X chromosome dosage and are not modulated by sex hormones [136]. Before brain exposure to gonadal secretions (i.e., prior to the critical period), cultured XX hypothalamic neurons exhibit a more accelerated differentiation than cultured XY neurons, in terms of neuritic arborization complexity and axonal length, with these sex differences being dependent on the bHLH neuritogenic factor *neurogenin 3* (*Ngn3*), which expression is increased in XX neurons [137,138]. *Kdm6a* is an upstream regulator of these sex differences in the neuritogenic development of hypothalamic neurons, expressing higher in XX than in XY neurons and being XX-specifically required for H3K27me3 demethylation at the *Ngn3* promoter region, upregulation of *Ngn3* expression and, subsequently, the promotion of axonal growth [136,139]. Importantly, among relevant proneural factors controlling hypothalamic neuronal differentiation, *Kdm6a* not only promotes the expression of *Ngn3*, but also of bHLH transcription factors *Ascl1* (*Mash1*), *Neurod1* and *Neurod2* and the neuron-specific activator of cyclin-dependent kinase 5 (CDK5) *Cdk5r1* (*p35*), albeit in a sex-independent manner in all these cases except for *Ngn3* [136,139]. Recently, it has been demonstrated that *Kdm6a* is also required for the expression of neuropeptides POMC and NPY, molecular markers of hypothalamic neuronal populations involved in the regulation of energy homeostasis and food intake. In sex-segregated hypothalamic neuronal cultures of E14 mice, *Pomc* and *Npy* showed sex-dependent and opposite expression patterns at both mRNA and protein levels, with female neurons expressing more *Pomc* and less *Npy* than male neurons. ChIP-qPCR data for H3K27me3 at *Pomc* and *Npy* promoter regions showed higher methylation levels in male than in female neurons, consistent with the higher expression of *Kdm6a* in females. *Kdm6a* knockdown induced an enrichment of H3K27me3 at *Pomc* and *Npy* promoters only in female neurons without affecting male neurons, showing that *Kdm6a* promotes chromatin accessibility at these loci specifically in females [139]. Together, evidence indicates that *Kdm6a* mediates these sex differences in *Pomc* and *Npy* expression by acting in females, not only through H3K27me3 demethylation at these loci, but also through activation of the bHLH *Ascl1*/*Ngn3* axis [139], both transcription factors that play a key role in regulating POMC+ and NPY+ cell fate specification in hypothalamus [140,141]. The effects of *Uty*, the Y homolog of *Kdm6a*, on these differentiation processes between XX and XY hypothalamic neurons have not been addressed in this model, so the question of whether *Uty*, which was conserved during Y evolution, might somehow be compensating for sex differences caused by *Kdm6a* dosage remains unanswered.

Loss-of-function mutations in *KDM6A* lead to Kabuki syndrome, a congenital disorder characterized by cognitive deficits, facial, skeletal and cardiac abnormalities and growth retardation [142,143,144]. As in many other X-linked diseases, male patients, who carry a single X and thus a single copy of *KDM6A*, tend to be more severely affected [145]. In a study assessing sex differences in vulnerability to Alzheimer’s disease, it was proposed that a second X chromosome in women confers resilience to the disease specifically through *KDM6A* [43]. Although many more women live with Alzheimer’s disease [146], largely due to their longer life expectancy, affected men die earlier and usually show more severe forms of the disease, with higher neurodegeneration and cognitive decline [147,148], despite similar β-amyloid (Aβ) and tau deposition between sexes [149,150,151]. XY neurons showed greater cell death than XX neurons when exposed in vitro to the neurotoxin Aβ, with modest *Kdm6a* overexpression reducing neurotoxicity in XY neurons and *Kdm6a* knockdown increasing it in XX neurons. Similarly, in mice engineered to express mutated forms of the human amyloid precursor protein (hAPP mice), individuals carrying a single X showed reduced longevity and worse spatial learning and memory performance compared to XX mice; when *Kdm6a* was overexpressed using lentivirus in the hippocampus of XY-hAPP mice, a significant improvement in learning and memory performance was observed in these animals compared to control XY-hAPP mice. Thus, the presence of a single X/*Kdm6a* copy consistently worsened hAPP/Aβ-related mortality, cognitive impairment and cellular viability compared to two X/*Kdm6a* copies. Finally, *KDM6A* expression in the human brain was higher in women than in men and in Alzheimer’s disease patients compared to controls. Considering all these observations in the human brain and in mouse models of the disease, authors suggested that having two copies of *Kdm6a* compared to just one copy of the gene confers stronger resilience to the disease, and speculated that increased *KDM6A* in brains of people with Alzheimer’s disease might be a neuroprotective, compensatory response [43].

KDM5C is another X-linked histone demethylase escaping XCI [81] and involved in sexual differentiation of the brain. Contrary to KDM6A, KDM5C activity contributes to repressed gene expression through H3K4 demethylation, with H3K4me3 being an epigenetic modification enriched in open, transcriptionally active chromatin sites [152,153]. Increased expression of *Kdm5c* in XX versus XY microglia has been associated with sex differences in microglial reactivity and microglia-mediated neuroinflammation after stroke in mice. After inducing ischemic stroke by middle cerebral artery occlusion, aged mice carrying two vs. one X chromosome, regardless of gonadal sex, showed larger infarct volumes, worse behavioral deficits, more robust microglial activation with increased pro-inflammatory TNFα and IL-1β and reduced anti-inflammatory IL-10 cytokines, and enhanced co-localization of KDM5C with the microglia marker IBA1 [154]. Remarkably, the expression of interferon regulatory factor 4 (IRF4), an important microglial anti-inflammatory element, was shown to be repressed by KDM5C H3K4 demethylation activity [155]. These results indicate an X dosage-dependent effect on stroke sensitivity, with microglial reactivity and microglia-mediated neuroinflammation after injury more exacerbated in XX vs. XY. Postmenopausal women show a greater vulnerability to stroke than men of the same age [156], so further research in this field could contribute to understanding whether sex chromosomes are involved and which X- and/or Y-linked genes regulate these sex differences. Interestingly, higher expression of *Kdm6a* in females has also been linked to neuroinflammation in mice, which might help shed light on women’s increased susceptibility to autoimmune diseases of the central nervous system such as multiple sclerosis. Using the experimental autoimmune encephalomyelitis (EAE) model of multiple sclerosis, which recapitulates CD4+ T cell-dependent disease, Itoh, et al. [157] showed that KDM6A deletion in CD4+ T cells ameliorated clinical disease, reduced neuropathology and downregulated the expression of neuroinflammation pathway genes, also providing an X dosage-dependent mechanism for sex differences in autoimmune diseases affecting the nervous system. Like KDM6A, KDM5C is also an important chromatin remodeler controlling transcriptional programs within neurons to impact their differentiation, neuritic growth and synaptic activity [158,159,160,161,162], with *KDM5C* mutations leading to Claes-Jensen X-linked intellectual disability [163,164,165], although further studies including sex as a critical variable are needed.

*Ddx3x* codes for an evolutionarily conserved DEAD-box RNA helicase that participates in numerous cellular events, such as mRNA synthesis, splicing, mRNA transport, translation and Wnt/β-catenin signaling pathway regulation [166,167,168]. Mutated forms of *Ddx3x* have been detected in different cancer types and are associated with intellectual disability (DDX3X syndrome), brain malformations, autism and epilepsy [108,169,170]. Most DDX3X syndrome individuals are female, an observation leading to the idea that the Y-linked homolog *DDX3Y* can compensate for *DDX3X* loss and contribute to milder phenotypes in males carrying *DDX3X* mutations [171,172]. Complete loss of *Ddx3x* in a conditional knockout (*Ddx3x*-cKO) mouse model led to microcephaly only in females, which might suggest that expression of the Y-linked homolog *Ddx3y* explains why *Ddx3x*-cKO male mice are phenotypically milder than *Ddx3x*-cKO females [173]. In addition, loss of one versus two *Ddx3x* copies caused vastly different corticogenesis phenotypes: *Ddx3x*-cKO females presented profound apoptosis in neural progenitors and neurons in the developing neocortex, whereas heterozygous females and *Ddx3x*-cKO males had impaired neurogenesis without cell death. *Ddx3x*-depleted progenitors exhibited prolonged cell division and more proliferative divisions at the expense of symmetric neurogenic divisions, affecting neural fates and reducing the generation of neurons. Finally, *Ddx3y* depletion phenocopies of cortical neurogenesis defects were observed in *Ddx3x*-cKO males, suggesting that *Ddx3y* can compensate for *Ddx3x* loss in males, an observation consistent with previous findings in both the hindbrain and hematopoietic system [174,175]. Similar results were reported by Lennox, et al. [176], with transient depletion of *Ddx3x* in the developing mouse neocortex disrupting corticogenesis by altering newborn cells distribution between cortical layers (more cells in the ventricular and subventricular zones and less in the cortical plate compared to control) and neuronal differentiation. Mechanistically, this impaired cortical neurogenesis after *Ddx3x* loss was suggested to be cause by defective helicase activity, formation of aberrant RNA-protein cytoplasmic granules and impaired translation of *DDX3X*-dependent mRNA targets observed in pathogenic *DDX3X* missense mutations [176].

## 5. Y-Linked Genes and Sexual Differentiation of the Brain

As mentioned above, X and Y chromosomes evolved from an autosomal pair and through a long process in which the original genetic information carried by the Y chromosome was severely reduced. Only about 3% of the ancestral genes originally located on the human Y chromosome still survive [177,178], in contrast to 98% of the ancestral genes on the X chromosome [179]. However, this loss/survival of Y genes did not occur randomly, but was subject to selection and strongly influenced by evolution of X and XCI. Evidence shows that most of the genes conserved on the modern Y have been retained due to selection because they belong to one of two categories: (1) genes playing male-specific functions in gametogenesis or reproductive development and (2) genes that are dosage-sensitive, broadly expressed X/Y homologs (mentioned above) [56,65,66]. Notably, surviving genes of the first category, such as *SRY*, *HSFY*, *TSPY* and *RBMY,* have shown a functional divergence from their X homologues (*SOX3*, *HSFX*, *TSPX* and *RBMX*, respectively) to control male reproduction. Nonetheless, because all Y genes have been exposed to selective pressure only in males, even widely expressed ancestral Y homologs may exhibit subtle differences in sequence and function from their X-linked pairs [55,56,66].

Contrary to the growing interest and research on X-linked genes in recent years, very little research has been carried out to delve into the role of specific Y-linked genes in the sexual differentiation of the brain. This is partly a consequence of the generalized belief that the Y chromosome encodes just a handful of protein-coding genes only important for male reproduction along with a much larger proportion of repetitive and/or noncoding “junk” genetic material. Although some studies have recently demonstrated, as discussed in the previous section, that expression of Y homologs of X/Y gene pairs such as *Uty*, *Ddx3y* and *Nlgn4y* is critical during brain development in males [117,133,173,174,180], ensuring a two-dose autosomal-like expression for these loci in XY that matches expression levels in XX (the latter carrying two active gene copies due to XCI escape), further research addressing the specific functions of Y-encoded homologs and differences with their X-linked counterparts at developmental and physiological levels is required. Of note, there has been recent interest in the Y noncoding genome and its contribution to both sex determination and other traits in health and disease [181]. For instance, at least six Y-linked long noncoding RNAs were found to be expressed in a time- and tissue-specific pattern in the developing brain of newborn chimpanzees, all sharing characteristics that suggest a potential functional role in brain development conserved in human and chimpanzee males [182]. Again, while quite intriguing and with great future potential, studies in this area are just beginning to yield results. Table 1 summarizes the X- and Y-linked genes discussed in this review and their main cellular functions.

## 6. Concluding Remarks

Today we know that the contribution of the X and Y chromosomes to the sexual differentiation of the brain and other non-gonadal tissues begins in the zygote itself and continues throughout life, via the differential expression of genes linked to these chromosomes per se and their influence on the expression of autosomal genes [58,88,89]. Importantly, both differentiating factors, i.e., sex chromosomes and gonadal hormones, can converge and interact on a given trait in a synergistic or antagonistic manner, establishing differences between the sexes or offsetting their effects and resulting in the attenuation/compensation of differences [183,184,185].

It took several decades of research and effort to comprehensibly identify sex chromosomes as causative factors of sex-specific phenotypes beyond sex/gonadal determination, and it was not until recent years that specific genes encoded by sex chromosomes began to be identified as agents of brain sexual differentiation. Due to the development of transgenic mouse models and powerful genetic tools to rigorously control the expression of particular genes in a time- and region-specific manner, both in vitro and in vivo, we are a little closer every day to a precise understanding of the action of X- and Y-linked genes in the brain. Hopefully, in the coming years we will be able to use this precise knowledge to address diseases with sex-biased phenomenology in the developing and aging brain.

## Figures and Tables

**Figure 1 ijms-23-12288-f001:**
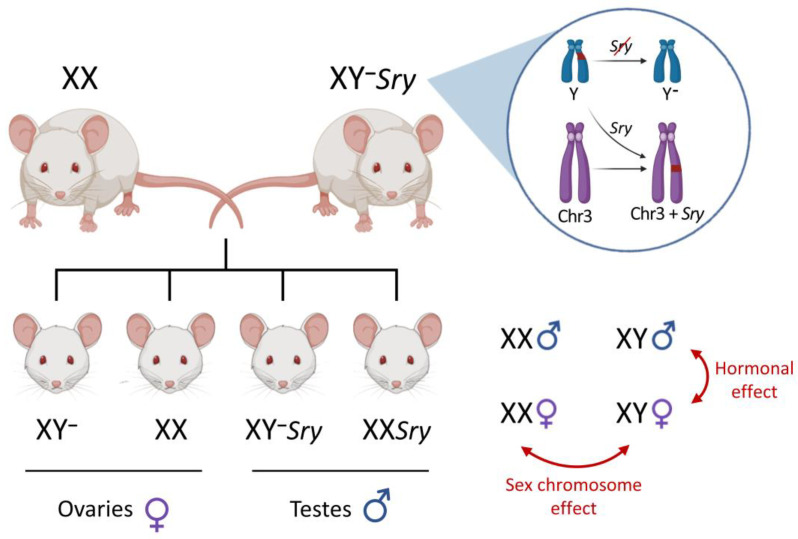
Four Core Genotypes (FCG) transgenic mouse model. XY^−^*Sry* mice carry two mutations that allowed to unlink the inheritance of the testis-determining *Sry* gene from that of the Y chromosome: (1) deletion of *Sry* from the Y chromosome, generating a “Y minus” or “Y^−^”; (2) reinsertion of *Sry* into chromosome 3 (Chr3). Breeding XY^−^*Sry* with XX mice makes it possible to obtain four different genotypes: XY^−^ and XX gonadal females (mice carrying ovaries), XY^−^*Sry* and XX*Sry* gonadal males (mice carrying testes). Comparison of a variable among the four genotypes allows independent assessment of the effects given by sex hormones (gonadal females vs. gonadal males), sex chromosome complement (XY vs. XX) or the interaction between these two factors (e.g., when a particular combination of gonadal type and sex chromosomes differs from the other three genotypes). ♀: gonadal female; ♂: gonadal male. Created with BioRender.com (accessed on 17 September 2022).

**Figure 2 ijms-23-12288-f002:**
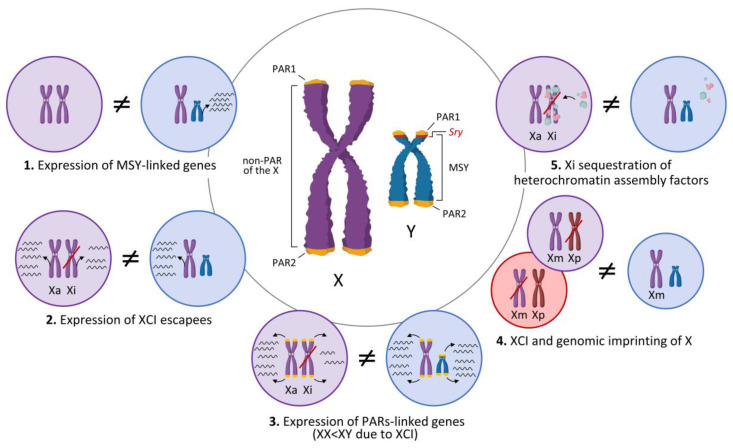
X and Y chromosomes and mechanisms of sexual differentiation. Representations of the human X and Y chromosomes (generalizable to all eutherian mammals) are presented in the center, showing the pseudoautosomal regions (PAR1 and PAR2) at the distal ends, the non-PAR of the X, the male specific region of the Y (MSY) and the *Sry* gene position. Five mechanisms of sexual differentiation generated by inequalities in genetic content, expression and inheritance of X and Y are shown (discussion in text). Xa: active X; Xi: inactive X; XCI: X chromosome inactivation; Xm: maternal X; Xp: paternal X. Created with BioRender.com (accessed on 17 September 2022).

**Table 1 ijms-23-12288-t001:** X- and Y-linked genes involved or potentially involved in brain sexual differentiation and their main functions. Source: Gene Ontology (GO) database http://geneontology.org/ (accessed on 8 October 2022).

Chr	Gene (GO Database)		
	Symbol	Name	Function
Y	*Sry*	sex determining region of Chr Y	DNA-binding transcription factor activity
	*Kdm6c-Uty*	histone demethylase UTY	Histone demethylase activity
	*Kdm5d*	lysine (K)-specific demethylase 5D	Histone demethylase activity
	*Ddx3y*	DEAD box helicase 3, Y-linked	ATP-dependent RNA helicase dbp3
	*Usp9y*	ubiquitin specific peptidase 9, Y chromosome	Cysteine-type deubiquitinase activity
	*Eif2s3y*	eukaryotic translation initiation factor 2, subunit 3, structural gene Y-linked	Eukaryotic translation initiation factor 2 subunit 3 family member
	*Nlgn4y*	neuroligin-4, Y-linked	Cell adhesion molecule
X	*Kdm6a-Utx*	lysine (K)-specific demethylase 6A	Histone demethylase activity
	*Kdm5c*	lysine (K)-specific demethylase 5C	Lysine-specific demethylase
	*Ddx3x*	DEAD box helicase 3, X-linked	ATP-dependent RNA helicase dbp3
	*Usp9x*	ubiquitin specific peptidase 9, X chromosome	Ubiquitin carboxyl-terminal hydrolase
	*Eif2s3x*	eukaryotic translation initiation factor 2, subunit 3, structural gene X-linked	Eukaryotic translation initiation factor 2 subunit 3 family member
	*Mecp2*	methyl CpG binding protein 2	DNA binding
	*Hdac8*	histone deacetylase 8	Histone deacetylase activity
	*Morf4l2*	mortality factor 4 like 2	Histone acetylation
	*Msl3*	MSL3 like 2	Histone acetylation
	*Phf6*	PHD finger protein 6	DNA metabolism protein
	*Fmr1*	fragile X messenger ribonucleoprotein 1	Regulation of alternative mRNA splicing
	*Mid1*	midline 1	E3 ubiquitin-protein ligase trim36-related
	*Ogt*	*O*-linked *N*-acetylglucosamine (GlcNAc) transferase (UDP-*N*-acetylglucosamine:polypeptide-*N*-acetylglucosaminyl transferase)	DNA binding
	*Syp*	synaptophysin	Lipid binding
	*Tmem47*	transmembrane protein 47	Plasma membrane

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
