# Peer review of "Genetics and Epigenetics of the X and Y Chromosomes in the Sexual Differentiation of the Brain"

_ijms, 2022, doi:10.3390/ijms232012288_

Round 1

Reviewer 1 Report

This review by Zapata et al. is well written and organized. Authors have comprehensively described the functional (epi)genetics of XY sex chromosomes and the association of gene contents to the brain functions and sexual differentiation. Authors present several cases from recent studies mainly from mammals highlighting the role of XY chromosome and evolutionary concepts involved in brain sexual differentiation.

I only have one minor comment for authors:

Can authors provide Gene ontology or a graph highlighting enriched functions for the X and Y linked genes (described in sections) to further support the hypothesis of this review?

Author Response

We appreciate the reviewer’s positive and constructive review of our manuscript. Following the reviewer’s request, we have built and included in the manuscript a table summarizing the discussed X- and Y-linked genes and their enriched functions stated by Gene Ontology (GO) database (http://geneontology.org/)

Reviewer 2 Report

This is a comprehensive review of a highly diverse literature, requiring a great deal of sophistication not only in the field of brain sexual differentiation, but also in the literature on the biology of sex chromosomes. There are very few scientists in the world with the breadth and acumen to write this kind of review.  The result is a highly useful review. Particularly valuable is the inclusion of a summary of the authors’ recent exciting work on the interaction of sex chromosome and gonadal hormonal factors.

The following are suggestions for consideration by the authors, to improve the manuscript. All of these are minor suggestions except those with an asterisk (*).

I suggest that the authors define male and female at the outset. This should become common practice, even in the biological literature, because of the increasingly diverse definitions that are used.

Line 66. The authors might wish to note that a feminizing role for estrogens in brain sexual differentiation is also indicated by studies of aromatase KO mice. Also, permanent “organizational” effects of androgens are seen also around puberty. I’m sure that the authors are fully aware that the theory described here has had a few revisions since originally proposed, and may have opted to present only the classic theory.

The manuscript is written in superb English.

Line 172. “Evidence” takes a singular verb or demonstrative pronoun. We do not say “evidences”.

*Line 174 and following. A subtle issue regards the phrase “the reported sex differences depend directly on the XY complement”. One could have equally correctly stated that the sex differences depend on the XX complement, but the implication would have been different. The phenotypic difference depends on a difference in effects of XX vs. XY sex chromosomes, not just the effects of one genotype or the other. Do the authors agree that a slightly better phrase might be “depend on the difference in XX-XY sex chromosome complement”?  By referencing only XY, or XX, there is the subtle implication regarding which complement actively makes one phenotype different from the other, which cannot be known from the evidence cited. The bias in this statement is clearer in line 176, where it is stated that male hormones have synergistic effects with male sex chromosomes. The interaction of sex chromosomes and sex hormones could actually involve an interaction between the XX genotype (e.g., when Kdm6a is high) with the developmental patterns caused when testosterone levels are low and/or ovarian hormones are high.

Line 178 typo “patterns”

Line 198. I suggest inserting “mammalian” before “X and Y chromosomes”.

Line 219. I do not believe that “many” of the Y genes retain an X-linked counterpart. In mouse at least, there are not many.

Line 228.  Typo, should be “800 code for proteins”.

Figure 2 and line 279. “Heterochromatic factors” is a bit misleading. These factors are not themselves heterochromatic, but they cause DNA to be heterochromatic. A better term would be “Heterochromatizing factors”.

Line 287. The Section 4 heading indicates that this section will discuss X-linked genes that are involved in brain sexual differentiation. Except for a few genes mentioned  only briefly at the outset, however, almost the entire discussion focuses entirely on the very small group of X gametologs of Y genes (Kdm6a, Kdm5c, Ddx3x). These genes have evolved the pattern of escape from X inactivation (higher expression in XX than XY cells in many tissues). I suggest that this section could be more impactful if the authors draw an analogy among these genes, pointing out their common characteristics, rather than listing them just as X genes. The title could begin the focus on this specific type of X gene. I think that would make this section stronger. However, the authors may have reasons not to accept this suggestion.

Line 302 and throughout the manuscript. Mouse and human proteins are all capitals, for example KDM5c, KDM6A. Mouse genes are italicized with first letter capitalized, Kdm5c.

Line 306 and nearby. Line 442. I recommend to say that a gene “codes for XYZ protein” or “encodes XYZ protein”, but not “encodes for”. Native speakers of English sometimes do not follow this advice, but I believe it to reflect proper English. It’s a minor point.

 Line 259 “before gonadal hormones organize…”

Paragraph 257. The authors review some of their very exciting recent work on Kdm6a here. I would be interested if they would comment on whether sex differences caused by Kdm6a might be mitigated by expression of Uty in XY cells. The effects of Uty in these cell systems have not been investigated, to my knowledge, but might they compensate for the sex differences caused by dose of Kdm6a?

Line 363. Please explain “opposite expression patterns” which could be opposite according to a number of variables mentioned here. 

*Line 385. I would urge caution here. The authors are claiming here that they have proven in ref 138 that Kmd6a regulates Pomc and Mpy expression by direct regulation of the promoter of these genes. Ref 138 is not available to me, but the information in the abstract does not reach this level of proof. If the authors have demonstrated that Kdm6a binds near these genes, that does not mean that such binding has a direct effect on the gene, although that finding is suggestive.  I urge the authors to state more precisely what was observed and can be concluded.

*Line 419. It is an overstatement to say that Kdm5c has been linked to stroke disease in humans. ”Linked” is a weak term, implying at most correlation between two variables, although it can unfortunately be mistakenly interpreted by the reader as a statement of cause and effect.  But there is no reported correlation between Kdm5c in microglia and disease in humans, to my knowledge. The authors should rephrase to indicate what evidence comes from mice, and how that evidence is used to speculate about human disease.

*Line 428. I do not think present evidence shows a “Kdm5c-mediated effect on stroke sensitivity” as stated by the authors, because the sensitivity to stroke has not been shown to vary with levels of Kdm5c expression.  Please rephrase to avoid this conclusion.

Author Response

We thank the reviewer for the thorough review on the manuscript. Following his/her suggestions and comments, we have edited the text to correct the indicated misspelled words, introduced some new passages, and discussed others in more detail, all of which we believe have significantly improved the revised manuscript.

Specific comments:

  • Regarding the organizational/activational classical hypothesis of brain sexual differentiation, we are aware of the reviewer’s observations on how it has been revised since it was originally proposed, but we opted to present only the classical theory for simplicity and since it is not the intended focus of the manuscript. Instead, we provide readers with excellent references where they can find the work of other authors discussing and delving deeper into the effects of gonadal hormones on the brain, if interested.
  • The term “heterochromatic factors” has been replaced by “heterochromatin assembly factors” in main text and Figure 2.
  • We have updated Ref 138 (now Ref 139), currently freely accessible in full on the journal website (Front Cell Dev Biol 2022, 10, 937875, doi:10.3389/fcell.2022.937875). Besides, we have revised and further discussed the section describing our latest published work on sex-dependent Kdm6a regulation of Ascl1, Ngn3, Pomc and Npy expression in hypothalamic neurons.